# Research Methods and Ethics in Health Emergency and Disaster Risk Management: The Result of the Kobe Expert Meeting

**DOI:** 10.3390/ijerph16050770

**Published:** 2019-03-03

**Authors:** Myo Nyein Aung, Virginia Murray, Ryoma Kayano

**Affiliations:** 1Advanced Health Science Institute and Faculty of International Liberal Arts, Juntendo University, Hongo 2-1-1, Bunkyo-ku, Tokyo 113-8421, Japan; myo@juntendo.ac.jp; 2Head of Global Disaster Risk Reduction, Public Health England, Wellington House, 133-155, Waterloo Road, London SE1 8UG, UK; 3World Health Organization Centre for Health Development, 1-5-1 Wakinohama-kaigandori, Chuo-ku, Kobe 651-0073, Japan; kayanor@who.int

**Keywords:** health emergency and disaster risk management (Health-EDRM), Sendai Framework for Disaster Risk Reduction, WHO Thematic Platform for Health-EDRM, research methods, ethics, glossary

## Abstract

In October 2018, at Asia Pacific Conference for Disaster Medicine (APCDM), an expert meeting to identify key research needs was organized by the World Health Organization (WHO) Centre for Health Development (WHO Kobe Centre (WKC)), convening the leading experts from Asia Pacific region, WHO, WHO Thematic Platform for Health Emergency and Disaster Risk Management (Health-EDRM) Research Network (TPRN), World Association for Disaster and Emergency Medicine (WADEM), in collaboration with Asia Pacific Conference for Disaster Medicine (APCDM) and Japan International Cooperation Agency (JICA). International experts, who were pre-informed about the meeting, contributed experience-based priority issues in Health-EDRM research, ethics, and scientific publication. Two moderators, experienced in multi-disciplinary research interacted with discussants to transcribe practical issues into related methodological and ethical issues. Each issue was addressed in order to progress research and scientific evidence in Health-EDRM. Further analysis of interactive dialogues revealed priorities for action, proposed mechanism to address these and identified recommendations. Thematic discussion uncovered five priority areas: (1) the need to harmonize Health-EDRM research with universal terms and, definitions via a glossary; (2) mechanisms to facilitate and speed up ethical review process; (3) increased community participation and stakeholder involvement in generating research ideas and in assessing impact evaluation; (4) development of reference materials such as possible consensus statements; and (5) the urgent need for a research methods resource textbook for Health-EDRM addressing these issues.

## 1. Introduction

At the Asia Pacific Conference for Disaster Medicine (APCDM) [1], October 2018, an expert meeting to identify key research needs in major research areas was organized by the World Health Organization (WHO) Centre for Health Development (WHO Kobe Centre (WKC), convening the leading experts from Asia Pacific region, WHO, WHO Thematic Platform for Health Emergency and Disaster Risk Management (Health-EDRM) Research Network (TPRN), World Association for Disaster and Emergency Medicine (WADEM). An expert meeting was conducted along with aseries of progresses in scientific aspects of the implementation of the 2015 Sendai Framework on Disaster Risk Reduction (SFDRR) [2], the resulting document of the Third UN World Conference on Disaster Risk Reduction (WCDRR), included the establishment of TPRN [3,4] and following journal papers on recommended Health-EDRM research activities [5,6]. Through the expert meeting and related review of literature and existing projects and activities, key research needs in five major Health-EDRM research areas were identified.

The Health-EDRM Network identified one major area of work that is important to address was clarity in relevant ‘Research Methods and Ethics’. The broad intersection of health and disaster risk reduction has resulted in an area of work now known as Health-EDRM which encompasses emergency and disaster medicine, disaster risk reduction, community health resilience, health system resilience, and impact of changing climate on health. Public health response during and after disasters has traditionally been focused on protecting populations from immediate threats [7]. Health-EDRM research involves the systematic analysis and management of health risks in emergencies and disasters by reducing the health risks and vulnerability. The complexity of undertaking research in disasters, and complying with ethical standards for these research, is critical but often much more difficult to ensure. This paper summarizes the outcome of the discussions and the proposed actions needed to support the delivery of Health-EDRM research.

## 2. Material and Methods

Prior to the meeting a range of international experts contributed experience-based priority issues in Health-EDRM research, ethics and scientific publication. It is of note that even in 1997, Stallings was able to state that “*… it is the context of research not the methods that makes disaster research unique*” [8]. The lead discussant, the rapporteur, and the other experts who participated to the discussion primarily aimed to identify priorities in scientific evidence on Research Methods and Ethics in Health Emergency and Disaster Risk Management Research focused on questions and issues to fill these gaps. In addition to identifying knowledge gaps, experts also aimed to assess knowledge-to-practice gaps in order to better integrate current expertise and research in this area into each phase of disaster risk management. Following a preliminary literature review and expert consultations, a series of questions that were thought to be very important to address for building better understanding of research methods and ethics in Health-EDRM included
(a)What are the definitions of research methods and technical terms for Health-EDRM?(b)How can impact evaluation methods for intervention and qualitative—quantitative mixed methods be standardized?(c)How can the publication process for Health-EDRM research become more systematic and effective?

Those participated in discussion were leading experts in Health-EDRM as well as country experts in disaster and emergency, and came from multiple disciplines such as public health, emergency medicine, nursing, and health care management. They discussed major issues in Health-EDRM research from practitioner viewpoints and different regional perspectives. Each issue was addressed by active discussion of participants with the aim of addressing priorities and actions on how to progress research and scientific evidence in Health DRM. Interactive dialogues noted simultaneously into minutes of discussion were further analyzed by the moderators into the gaps, proposed mechanism to overcome the gaps and to provide a summary of recommendations.

## 3. Results and Discussion

Experts from different parts of the globe participated in this thematic group discussion. The findings from this discussion were wide ranging. It was noted that research findings from disaster research is not easily translated into different contexts of the many countries around the world, and there were challenges and difficulties in implementing practices in addressing national health system, cultural and religious issues before, during, and after interventions. These issues were difficult to address without more complete and systematic evidence to inform Health-EDRM policy and practice. Although translating research findings into policy is the ambition of many researchers and practitioners in order to develop evidence-based policies, there are issues of how much researchers can communicate with policy makers in comparison to the opportunities to facilitate policy makers’ uptake of research findings. A key strategy to overcome this barrier is stakeholders’ involvement and community participation since the development of research ideas in designing phase of the research project. For example, research in disaster affected area might recruit participants who were disasters casualties and there should be rules and regulation especially listing ethical ‘don’t’s such as providing food as an incentive for participation in the research. For instance, food is sometimes used as incentive for the participation in the research, in some context. However, in the disaster and emergency situation, food is the basic need provided as the humanitarian aid regardless of participation in the research. Thus, if food is used as incentives for research recruiting such persons, it might be forcing someone to participate in the research, in the fear that they cannot receive food supply. Therefore, it is not appropriate to use food as incentives in Health-EDRM research.

Thematic discussion in the meeting uncovered five priority areas: (1) the need to harmonize Health-EDRM research with universal terms and definitions via a glossary; (2) mechanisms to facilitate and speed up ethical review process; (3) development of reference materials such as possible consensus statements; (4) increased community participation and stakeholder involvement in generating research ideas and in assessing impact evaluation; and (5) and the urgent need for a textbook for Health-EDRM research addressing these and other issues. Discussions also agreed that there was a need to harmonize definitions in Health-EDRM research with universal terms, and the development of a glossary of definitions. Such a glossary could promote common understanding and common usage of concepts, terms, and aims for Health-EDRM. If undertaken by WHO, the glossary might have the additional advantage of being translated into the WHO official languages. Even though the United Nations Office for Disaster Risk Reduction (UNISDR) convened an Open-ended Intergovernmental Expert Working Group (OIEWG) to report on indicators with recommended terminology relating to disaster risk reduction, which was delivered in 2016 [9], and adopted by the UN General Assembly on 2 February 2017 and updated the 2009 UNISDR Terminology on Disaster Risk Reduction [10]; however, not all the necessary Health-EDRM terms were included in these or other terminologies.

The need to find mechanisms to facilitate and speed up ethical review process was addressed and the need to quicken the review process for disaster research ethics was noted as being complex and very dynamic. A basic requirement was to agree on methodological terms and should reflect research undertaken in the spectrum of disaster chronology such from preparedness and risk reduction, to emergency management, and to post disaster rehabilitation. One key suggestion was the requirement to reduce the review time for ethical approval with, wherever possible, with pre-agreed ethical approval. As Chan et al. (2019) pointed out their recent Lancet editorial that “research stakeholders have a responsibility to protect the interests of communities involved in research, achieving this is rarely straightforward in emergencies” [11]. From the discussion that it would require international consensus among the professionals and researchers, possibly using as good practice models the CONSORT (Consolidated Standards of Reporting Trials) statement for randomized controlled trials [12,13] and STROBE (STrengthening the Reporting of OBservational studies in Epidemiology) statement for cohort studies [14,15]. Additional models of good practice such as the Guidance for Managing Ethical Issues in Infectious Disease Outbreaks [16] and the Health Emergency and Disaster Risk Management fact sheet on ETHICS [17] were thought to be practical and helpful examples. More data on these topics to address Health-EDRM were considered to be of importance. Once a research proposal can fit in a standardized check list, the ethical committee should be able to agree the research proposed via an expedited channel.

Such activities could be part of the development of reference materials such as possible consensus statements. Such consensus statement provide a checklist of methodology details which allows researcher to check the proposal themselves. Ethical committee can quickly assessthe quality assurance and safety through the list, speeding up the approval. Therefore, it is very essential step to develop a consensus for disaster research.

The call for increased community participation and stakeholder involvement in generating research ideas and in assessing impact evaluation was considered and the need to listen to the voice of the affected and their community leaders and local representatives is increasingly critical. However, relatively few Health-EDRM reports on community participation and stakeholder involvement in generating research ideas were shared in the discussion. There are examples of where research to address stakeholder involvement in health research, such as the recent report from Kapiriri (2018) [18], but it is not from emergency or disaster research. Very little has been published definitively on this area.

Much more is described in the quest of methodological rigour for impact evaluation. The experts brain-stormed how can impact evaluation apply quantitative methods in additional to commonly applied case-studies in disaster research.

Impact evaluation is usually applied to see how a research programme works well in a particular setting. It is a term understood readily by multiple stakeholders whereas it dictates how to measure the outcome an intervention in basic research methodology. There are many methodological tools to measure how an intervention brought about changes in comparison to pre-existing situation or in comparison to naïve control group. A proposed intervention should be tested by efficacy and effectiveness trials before it comes into the guidelines and practice. It was considered that the synthesis of evidence would be primarily through the process of systematic reviewing and, if appropriate, modelling and cost effectiveness decision analysis [19]. The efficacy depends on how the intervention is planned to measure (design), how accurate are the measurement tools (validity and reliability) whereas the effectiveness will inform how robust is the intervention in the real world setting. Furthermore, the scalability of intervention will be challenged by the economic, social, and political context and the resource need. It would be important to follow process recommended where possible by organisations and their reports of activities that are already engaged in working in this area such as WHO [20,21], UK Medical Research Council [22], Organisation for Economic Co-operation and Development [23], and the Humanitarian Policy Group at the Overseas Development Institute and its Good Practice Review [24] with its chapter on Monitoring and Evaluation [25,26].

Diverse research methods from case studies, natural experiment designs and randomized controlled trials can be applied in impact monitoring and evaluation. For example, clinical epidemiology is a discipline to determine clinical outcomes applying tools of epidemiology. Likewise, epidemiological tools can be selected to match where and how they would be applied in Health-EDRM research but are often less easy to monitor and evaluate as they are used in more difficult environments. Through the synergy of Health-EDRM experts and the discussion on research methods, designs, and tools, it might be possible to encourage selection and use of more appropriate mechanisms. Events to gather such multidisciplinary professionals such as international workshops are important opportunity to generate the list of strategic research methodology tools. Relatively few resources for disaster risk reduction and management research methods have been identified and some excellent examples are cited below [27,28,29,30,31,32]. However most of these do not specifically address the full range of Health-EDRM research domains. Therefore, it is considered that there is relatively little currently that reflects the wide needs of Health-EDRM researchers, practitioners, and policy makers. Therefore, a text book linked to a website for easy updating is needed and such a resource could be tentatively entitled “research methods for health emergency and disaster risk management”. This would be a rewarding endeavor and would be beneficial to the establishment of the WHO Thematic Platform for Health Emergency and Disaster Risk Management Research Network.

## 4. Conclusions

It is hoped this paper on key issues in research methods and ethics in health emergency and disaster risk management will contribute to the identification and implementation of concrete solutions that foster the creation and the use of knowledge for research and ethics building resilience before, during, and after disasters. In the discussion, issues were raised on potential gap areas in the disaster research methodology: impact evaluations; Consensus on the definitions; development of common research statements like the CONSORT or STROBE for disaster research; how research findings in disaster research can be translated into different contexts of the many countries around the world; and the need to develop the textbook in disaster research methodology to help guide international researchers.

As for other major Health-EDRM research areas, research and ethics requires collaboration between experts, decision-makers, practitioners, and communities in order to facilitate coordinated response when and where it is most needed. It is critical a text book is created to provide a reference which would be fundamental and global contribution to the establishment of the WHO Thematic Platform for Health Emergency and Disaster Risk Management Research Network.

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
