# Peer review of "Research Methods and Ethics in Health Emergency and Disaster Risk Management: The Result of the Kobe Expert Meeting"

_ijerph, 2019, doi:10.3390/ijerph16050770_

Round 1
Reviewer 1 Report
Thank you for inviting me to review this interesting and meaningful piece of work. Overall the authors are able to communicate the essence of the discussion from the meeting. However, the grammatical issue must be addressed so that the readers can understand the manuscript better.
- The paper will benefit from more description of the examples raised in the meeting, e.g.
Page 3 line 102:
"For example, research in disaster affected area might recruit participants who were disasters casualties and there should be rules and regulation especially listing ethical ”don’ts” such as providing food as an incentive for participation in the research."
Is it that providing incentives is an issue in the disaster setting? or is it that the use of food as an incentive?
- Some of the grammatical issues as follows:
Page 1 line 26: five
Page 1 line 27: speed up
Page 2 line 83: how
Page 2 line 85: full stop at the end of the sentence
Page 3 line 99: the development of
Page 3 line 105: speed up
Page 3 line 112: might have
Page 3 line 117: “…February 2nd, 2017, It updated of the publication” full stop. It has updated on
Page 3 line 121: speed up
Page 3 line 123: agree on
Page 3 line 126: with pre-agreed
Page 4 line 136: of importance
Page 4 line 167: real world
Page 4 line 168: furthermore
Page 5 line 201: CONSORT, STROBE
Author Response
Thank you very much for your time and contribution to edit our manuscript and your impression.
Reviewer's comment
- Some of the grammatical issues as follows:
Page 1 line 26: five
Page 1 line 27: speed up
Page 2 line 83: how
Page 2 line 85: full stop at the end of the sentence
Page 3 line 99: the development of
Page 3 line 105: speed up
Page 3 line 112: might have
Page 3 line 117: “…February 2nd, 2017, It updated of the publication” full stop. It has updated on
Page 3 line 121: speed up
Page 3 line 123: agree on
Page 3 line 126: with pre-agreed
Page 4 line 136: of importance
Page 4 line 167: real world
Page 4 line 168: furthermore
Page 5 line 201: CONSORT, STROBE
Author’s reply
Thank you very much for your kind review. All the proposed suggestions and corrections are reflected on the revised version.
Reviewer's comment
- The paper will benefit from more description of the examples raised in the meeting, e.g.
Page 3 line 102:
"For example, research in disaster affected area might recruit participants who were disasters casualties and there should be rules and regulation especially listing ethical” don’ts” such as providing food as an incentive for participation in the research." Is it that providing incentives is an issue in the disaster setting? or is it that the use of food as an incentive?
Author’s reply
For instance, food is sometimes used as incentive for the participation in the research, in some context. However, in the disaster and emergency situation, food is the basic need provided as the humanitarian aid regardless of participation in the research. Thus, if food is used as incentive for research recruiting such persons, it might be forcing someone to participate in the research, in the fear that they cannot receive food supply. Therefore, it is not appropriate to use food as incentives in Health-EDRM research.
This point was addressed in the expert meeting. According to the discussion, we added some more sentences to make the statement more understandable. We have added examples in relevant such as CONSORT or STROBE statement, pre-agreed ethical approval. Since there is limited space for a short communication, I hope editor may understand us.
Thank you very much.
Authors

Reviewer 2 Report
Substantively, the content of the article has merit. The methodology could benefit from a clearer description, and the results of the expert debate are valuable.
Some of the resources promoted (26- 29) should be investigated in greater depth, although the word Method or Methodology is used in the title, they are NOT research methods books. I am curious as to why Social Science Research Methodology and Public Health Research Methods textbooks not considered in the query and overview. Further, an important document to integrate into the discussion is the IRDR Forensic Investigation of Disasters: A Conceptual Framework and Guide to Research. The pdf can be downloaded at: http://www.irdrinternational.org/2016/01/21/irdr-publishes-the-forin-project-a-conceptual-framework-and-guide-to-research/
The numerous writing errors made it difficult to assess the substance of the paper. Noted writing errors to correct are attached.
Author Response
Thank you very much for your time and contribution to edit our manuscript and your impression. We do thank you for the valued comments which improved the manuscript.
We have added recommended references
Reference 28
Oliver-Smith, A., Alcántara-Ayala, I., Burton, I., & Lavell, A. (2016). Forensic investigations of disasters (FORIN): A conceptual framework and guide to research. IRDR FORIN Publication
Reference 32
Oakes, J. M., & Kaufman, J. S. (Eds.). (2017). Methods in social epidemiology (Vol. 16). John Wiley & Sons.
Reference 27, 29 and 30 are also textbooks. Writing errors are carefully checked and corrected thanks to another reviewer.
